# Cervical and Vaginal Microbiomes in Early Miscarriages and Ongoing Pregnancy with and without Dydrogesterone Usage

**DOI:** 10.3390/ijms241813836

**Published:** 2023-09-08

**Authors:** Mariya Gryaznova, Olesya Kozarenko, Yuliya Smirnova, Inna Burakova, Mikhail Syromyatnikov, Alexander Maslov, Olga Lebedeva

**Affiliations:** 1Laboratory of Metagenomics and Food Biotechnology, Voronezh State University of Engineering Technologies, 394036 Voronezh, Russia; mariya-vg@mail.ru (M.G.); kozarenko.o@vsuet.ru (O.K.); dyd16@mail.ru (Y.S.); vitkalovai@inbox.ru (I.B.); syromyatnikov@bio.vsu.ru (M.S.); alex.maslov@einsteinmed.edu (A.M.); 2Antenatal Care Department, Yakovlevo Central District Hospital, 309070 Stroitel, Russia; 3Department of Genetics, Cytology and Bioengineering, Voronezh State University, 394018 Voronezh, Russia; 4Department of Genetics, Albert Einstein College of Medicine, New York, NY 10461, USA; 5Department of Obstetrics and Gynecology, Belgorod State National University, 308015 Belgorod, Russia

**Keywords:** cervical microbiome, vaginal microbiome, early pregnancy, miscarriage, dydrogesterone, *16S rRNA*, NGS, *Lactobacillus iners*, *Gardnerella vaginalis*, *Mycoplasma*, *Bifidobacterium*

## Abstract

Emerging evidence suggests that the reproductive tract microbiota is a key modulator of local inflammatory and immune pathways throughout pregnancy and may subsequently impact pregnancy outcomes. In this study, our objective was to analyze the cervical and vaginal microbiomes during early pregnancy among three groups: women with healthy ongoing pregnancies, women undergoing dydrogesterone treatment, and those who experienced miscarriages. The experiment involved 51 women at 8–11 weeks of gestation. The microbiome was examined using *16S rRNA* sequencing on the Ion Torrent PGM platform. Across all groups, *Lactobacillus iners* was predominant, suggesting that the vaginal community type CST III is common among the majority of participants. Notably, our data highlighted the significant roles of *Gardnerella vaginalis* and *Mycoplasma girerdii* in the pathogenesis of early miscarriage. Conversely, *L. iners* and *Bifidobacterium longum* have a protective effect in early pregnancy. Moreover, dydrogesterone intake appeared to influence notable differences between the cervical and vaginal microbiomes. Overall, our study enhanced our understanding of the cervical and vaginal microbiome composition in the eastern European population during early pregnancy.

## 1. Introduction

The rate of early miscarriages in the general population is about 15% among clinically recognized pregnancies [1]. Early miscarriage, especially recurrent pregnancy loss, can lead to several medical (sepsis, bleeding, infertility) and psychological (depression, anxiety) conditions and can be a risk factor for obstetrical complications during the next pregnancy [2]. Meanwhile, the role of the genital tract microbiome in the pathogenesis of miscarriages is poorly understood. 

Overall, *Lactobacillus*-dominated types of the microbiome were found to be more beneficial than non-*Lactobacillus*-dominated types [3,4]. However, the data on less abundant microorganisms in the microbiome of patients with early miscarriages differ among the studies. It can be because the genital tract microbiome composition differs depending on ethnicity. The review by Gupta et al. (2017), summarizing the data on the vaginal microbiome in non-pregnant women, indicated that women of European and Asian ancestry have mostly *Lactobacillus*-dominated types of the microbiome, while African and Latin American women mostly have microbiomes dominated by anaerobic microorganisms associated with bacterial vaginosis [5]. These differences might be explained not only by diet and lifestyle but also by genetic polymorphisms in local immunity factors [6]. 

Differences in microbiome composition persist during a normal early pregnancy. Thus, in Chinese, eastern European, Caribbean, and United States populations, the predominant microorganism during the first trimester of pregnancy is *Lactobacillus iners* [7,8,9,10,11,12], which corresponds to the community state type III (CST III) microbiome according to the classification by France et al. (2020) [13]. In Canadian and British populations, the predominant microorganism of the lower genital tract in early pregnancy is *L. crispatus*, which corresponds to CST I [14,15]. 

Meanwhile, to our knowledge, there are no data on the microbiome composition of the eastern European population in patients with early miscarriages. Another problem that has not been studied before is the microbiome composition of patients using progestins during early pregnancy. Nowadays, progestins are the main medications used for the treatment of threatened miscarriages in the first trimester in all international guidelines. Similarly, to cases with “near-missed” women, which can give a lot of information about the prevention of maternal mortality, ongoing pregnancies with progesterone usage can be considered “near-missed pregnancies”, and such patients might have the genital tract microbiome, which can differ from ongoing pregnancy without the progestin usage. Therefore, the comparison of cervical and vaginal microbiomes in women with ongoing pregnancies with and without progestin usage, as well as with early pregnancy loss, is of great interest. 

Thus, the purpose of this study is to determine the composition of the microbiome of the lower genital tract of three groups of women: women with an ongoing pregnancy who did not take dydrogesterone, women with an ongoing pregnancy who took dydrogesterone, and women with miscarriage, to determine the critical components of their microbiomes and look for patterns that are characteristic of these pregnancy states using the next-generation sequencing platform (NGS).

## 2. Results

No significant differences between groups were observed in clinical characteristics, except for the higher rate of miscarriages in the history of patients in groups II and III compared to group I (Table 1). The rate of previous miscarriages was not different between patients in groups II and III.

### 2.1. Microbiome Composition of the Cervical Canal and Vagina

In total, 120 bacterial species were identified while studying the cervical and vaginal microbiomes of patients with ongoing pregnancies and miscarriages. However, the abundance of most of the identified bacteria was extremely low, as a result of which we combined the least represented species into the “Other” group. Figure 1 and Figure 2 show the most abundant bacterial species of the cervical (Figure 1) and vaginal (Figure 2) microbiomes, whose numbers exceeded 1% for at least one studied group.

A total of ten bacteria formed the central microbiome core in the cervical canal. *Lactobacillus iners* was the most abundant species for all groups. So in the group of patients with ongoing pregnancy who did not take dydrogesterone, the number of *Lactobacillus iners* was 54.17%; in those who took dydrogesterone, it was 47.84%; and in patients with miscarriage, it was 47.96%. The next abundant bacterium for all groups was *Gardnerella vaginalis*. In the group of patients with ongoing pregnancy who did not take dydrogesterone, its number was 20.62%; in the group taking dydrogesterone, it was 9.99%; and in the group with miscarriage, it was 25.23%. The remaining bacteria were distributed differently in the studied groups.

Thus, in patients with ongoing pregnancy who did not take dydrogesterone, the next largest after *Lactobacillus iners* and *Gardnerella vaginalis* was *Bifidobacterium longum*, whose number was 5.23%. Then, the bacteria were distributed as follows: *Prevotella copri* (1.43%); *Bifidobacterium dentium* (1.11%); *Ileibacterium valens* (1.07%); *Pseudomonas aeruginosa* (0.81%); *Mycoplasma girerdii* (0.27%); *Mycoplasma hominis* (0.11%); and *Brochothrix thermosphacta* (0.09%). 

In the group of patients with ongoing pregnancy who took dydrogesterone, the following were found in abundance: *Bifidobacterium dentium* (5.62%); *Prevotella copri* (4.09%); *Brochothrix thermosphacta* (2.85%); *Bifidobacterium longum* (2.84%); *Ileibacterium valens* (2.83%); *Pseudomonas aeruginosa* (2.05%); *Mycoplasma hominis* (0.09%); and *Mycoplasma girerdii* (0.06%).

In the group of patients with miscarriage, the next most abundant species were as follows: *Mycoplasma girerdii* (3.86%); *Bifidobacterium dentium* (2.93%); *Mycoplasma hominis* (2.79%); and *Prevotella copri* (1.96%).

At the same time, 12 bacteria formed the core of the vaginal microbiota. As well as in the cervical microbiome, *Lactobacillus iners* was the most abundant bacterium in the vaginal microbiome. We observed its largest number in the group of patients with ongoing pregnancy who did not take dydrogesterone (62.97%); a slightly smaller number was observed in the group of patients with ongoing pregnancy who were taking dydrogesterone (54.67%); and the smallest number was in the group with miscarriage (47.01%). *Gardnerella vaginalis* was the next abundant bacterium in all groups. In patients with ongoing pregnancy who did not take dydrogesterone, the number was 14.40%; in patients taking dydrogesterone, it was 9.78%; and in patients with miscarriage, we observed the highest number (26.18%). The remaining bacteria were distributed differently in the studied groups.

In the group of patients with ongoing pregnancy who did not take dydrogesterone, the next abundant species were as follows: *Bifidobacterium longum* (4.20%); *Lactobacillus gasseri* (3.18%); *Ileibacterium valens* (1.62%); *Prevotella copri* (1.33%); *Bifidobacterium dentium* (1.14%); *Mycoplasma girerdii* (0.34%); *Mycoplasma hominis* (0.23%); *Bacteroides plebeius* (0.15%); *Bifidobacterium breve* (0.10%); and *Bacteroides massiliensis* (0.03%).

In the group of patients with ongoing pregnancy who took dydrogesterone, the following were found in abundance: *Lactobacillus gasseri* (6.82%); *Bifidobacterium dentium* (5.96%); *Ileibacterium valens* (2.83%); *Bacteroides plebeius* (1.90%); *Prevotella copri* (1.56%); *Bifidobacterium longum* (1.21%); *Bacteroides massiliensis* (0.48%); *Mycoplasma girerdii* (0.32%); *Mycoplasma hominis* (0.16%); and *Bifidobacterium breve* (0.09%). 

In the group of patients with miscarriage, the next most abundant species were as follows: *Mycoplasma girerdii* (4.60%); *Bacteroides plebeius* (4.41%); *Bifidobacterium dentium* (2.78%); *Mycoplasma hominis* (2.06%); *Bifidobacterium breve* (1.23%); *Bacteroides massiliensis* (1.02%); and *Prevotella copri* (0.85%).

### 2.2. Indicators of Intragroup and Intergroup Microbiome Diversity

We also assessed both the intergroup and intragroup diversity of microbiomes. We used the Shannon index to evaluate alpha diversity, and beta diversity was assessed using the Bray–Curtis dissimilarity metric. Figure 3 and Figure 4 show the diversity analysis results for the cervical and vaginal microbiomes of the patients.

The group clustering of the cervical microbiome between the study groups was not identified. Regarding intragroup diversity, the cervical microbiome of patients with miscarriage was the richest and most diverse (H = 0.93), and it also statistically significantly exceeded the diversity in the groups with ongoing pregnancy in patients who did not take dydrogesterone (H = 0.49, *p-value* = 0.0007) and took it (H = 0.37, *p-value* = 4.78 × 10^−6^). There were no statistically significant differences in the alpha diversity of the cervical microbiome between the groups with ongoing pregnancy who took and did not take dydrogesterone.

The most diverse vaginal mycobiome was also characteristic of the miscarriage group (H = 1.20); it was statistically significantly different from the group with ongoing pregnancy, in which the patients did not take dydrogesterone (H = 0.43, *p-value* = 2.33 × 10^−8^), and from the group that took dydrogesterone (H = 0.60, *p-value* = 0.0004). Intergroup clustering of microbiomes, which would reflect the presence of beta diversity, was not observed.

### 2.3. Differential Abundance of Bacteria

We did not find any difference in cervical and vaginal CSTs according to the classification by France et al. (2020), which is based on the species of dominant microorganisms [13] (Appendix A). Meanwhile, the abundance of bacterial species showed significant differences between groups. 

Analysis of the differential abundance of the microbiome of the cervical canal in patients from the study groups revealed statistically significant differences for several bacteria (Figure 5).

Thus, the highest abundance of *Bifidobacterium dentium* was found in the cervical microbiome of patients with ongoing pregnancy taking dydrogesterone (5.62%). The number of *Bifidobacterium dentium* in this group was statistically significantly higher than the number in the group of patients with ongoing pregnancy who did not take dydrogesterone (1.50%; *p-value* = 0.004), as well as in the group of patients with miscarriage (2.93%; *p-value* = 0.006).

*Bifidobacterium longum* was not represented in the group of patients with miscarriage and was statistically different from the group with ongoing pregnancy, in which patients did not take dydrogesterone (4.55%; *p-value* = 0.005), and from the group with ongoing pregnancy, where women took dydrogesterone (2.84%; *p-value* = 0.01).

The abundance of *Dialister invisus* was statistically significantly different between the group with miscarriage, in which this bacterium was absent, and the group with ongoing pregnancy, where dydrogesterone was taken (1.08%; *p-value* = 0.02).

The abundance of *Gardnerella vaginalis* was statistically different between all studied groups, in particular between the groups with ongoing pregnancy, where the patients did not take dydrogesterone (18.89%), and the group with ongoing pregnancy, where they took dydrogesterone (9.99%; *p-value* = 0.0001). The same was true between the miscarriage group (25.23%) and the group with ongoing pregnancy without taking dydrogesterone (*p-value* = 0.0001), and between the group with miscarriage and the group with ongoing pregnancy where dydrogesterone was taken (*p-value* = 0.0001).

The species *Lactobacillus iners* was the most abundant in the group with ongoing pregnancy, in which the patients did not take dydrogesterone (56.53%), and statistically significantly differed from both the group of pregnant patients taking dydrogesterone (47.84%; *p-value* = 0.0001) and from the group of patients with miscarriage (47.96%; *p-value* = 0.0001).

The greatest number of *Mycoplasma girerdii* was observed in the group of patients with miscarriage (3.86%), and it was statistically different from the abundance of this bacterium in the group of patients with ongoing pregnancy who took dydrogesterone (0.06%; *p-value* = 0.03) and did not take dydrogesterone (0.55%; *p-value* = 0.04).

The abundance of *Prevotella copri* was statistically different between the groups with ongoing pregnancy, where the patients took dydrogesterone (4.09%), and the group where they did not take it (1.24%; *p-value* = 0.04).

We also analyzed the quantitative composition of the vaginal microbiome (Figure 6).

For the vaginal microbiota, differential abundance analysis revealed differences in ten bacteria between study groups. *Bacteroides plebeius* prevailed in the group of patients with miscarriage (4.41%), and its abundance was statistically significantly different from that in the group of patients with ongoing pregnancy who did not take dydrogesterone (0.21%; *p-value* = 0.01).

The abundance of *Bifidobacterium breve* in the group of patients with miscarriage (1.23%) was statistically significantly different both from the group of patients with ongoing pregnancy who did not take dydrogesterone (0.12%; *p-value* = 0.01) and from the group where the patients took dydrogesterone (0.09%; *p-value* = 0.04).

For *Bifidobacterium dentium*, we observed statistical differences between the group of patients with ongoing pregnancy taking dydrogesterone (5.96%) and the other two study groups: the group of patients with ongoing pregnancy who did not take dydrogesterone (1.54%; *p-value* = 0.003) and the group of patients with miscarriage (2.78%; *p-value* = 0.008).

*Bifidobacterium longum* was the most numerous in patients with ongoing pregnancies who did not take dydrogesterone (4.86%). Its number was statistically significantly different both from the group of patients with ongoing pregnancy who took dydrogesterone (1.21%; *p-value* = 0.01) and from the group of patients with miscarriage (0.00%; *p-value* = 0.005).

The abundance of *Gardnerella vaginalis* was statistically different between all studied groups, as well as in the study of the microbiomes of the cervical canal. Between the groups with ongoing pregnancy, where the patients did not take dydrogesterone (18.60%), and the group with ongoing pregnancy, where they took dydrogesterone (9.78%; *p-value* = 0.0001). The same was true between the miscarriage group (26.18%) and the group with ongoing pregnancy without taking dydrogesterone (*p-value* = 0.0001), and between the group with miscarriage and the group with ongoing pregnancy where dydrogesterone was taken (*p-value* = 0.0001).

For *Lactobacillus gasseri*, we observed statistical differences between the group of patients with ongoing pregnancy taking dydrogesterone (6.82%) and the other two study groups: the group of patients with ongoing pregnancy who did not take dydrogesterone (0.28%; *p-value* = 0.0001) and the group of patients with miscarriage (0.00%; *p-value* = 0.0002).

*Lactobacillus iners* was the most abundant in the group with ongoing pregnancy, in which the patients did not take dydrogesterone (64.65%), and statistically significantly differed from both the group of pregnant patients taking dydrogesterone (54.67%; *p-value* = 0.0001) and from the group of patients with miscarriage (47.01%; *p-value* = 0.0001). The abundance of *Lactobacillus iners* also significantly differed between the group of patients with ongoing pregnancy, in which dydrogesterone was taken, and the group with miscarriage (*p-value* = 0.0001). The greatest number of *Mycoplasma girerdii* was observed in the group of patients with miscarriage (4.60%), and it was statistically different from the group of patients with ongoing pregnancy who did not take dydrogesterone (0.46%; *p-value* = 0.02) and took dydrogesterone (0.32%; *p-value* = 0.02).

## 3. Discussion

In this study, we determined the microbiome of the vagina and cervix in patients in early pregnancy as well as in patients with miscarriages. Thus, we had three study groups, dividing the group with ongoing pregnancy into one where women took dydrogesterone and one where women did not take this medication. Among these study groups, we identified differences in the microbiome composition of the cervix and vagina and also determined their alpha diversity and the degree of intergroup similarity.

We did not find a significant difference in CSTs between study groups according to the classification of France et al. (2020) [13]. In all groups, we observed the predominance of *Lactobacillus iners* in the microbiome of both the vagina and the cervical canal. This suggests that the majority of our patients are characterized by the type of vaginal community CST III [13]. Meanwhile, the results of our study showed a significantly higher abundance of *L. iners* in the group of patients with ongoing pregnancy who did not take dydrogesterone compared with the other two study groups.

As stated before, *L. iners* was found to be the predominant bacteria during the first trimester of normal pregnancy in eastern European, Chinese, Caribbean, and United States populations [7,8,9,10,11,12], and second after *L. crispatus* in British and Canadian populations [14,15]. According to large research studies, there are no changes in the CSTs across trimesters of pregnancy [11,14]. 

In the non-pregnant state, *L. iners* is the predominant species in the vaginal microbiota among older women and women of African American descent. *L. iners* is also frequently isolated from the vagina of women diagnosed with bacterial vaginosis, shortly after treatment and during menstruation [16]. This species is supposed to be very flexible and can quickly adapt to changing vaginal conditions. Functional analysis of proteins encoded by *L. iners* showed that this bacterium can exhibit both commensal and pathogenic properties [17,18].

According to recent studies, the predominance of *L. iners* in the vagina of pregnant women in the first trimester increases the risk of preterm labor and miscarriage compared to the predominance of *L. crispatus* [19,20,21]. Meanwhile, in our study, the prevalence of *L. iners* had a protective effect. 

In the vaginal microbiome, we observed that the abundance of *Lactobacillus gasseri* was significantly higher in the group of patients with ongoing pregnancy treated with dydrogesterone compared to pregnant patients who did not take it, as well as in the miscarriage group. *L. gasseri* defines the CST II vaginal community type and can produce lactic acid and hydrogen peroxide, acidify the vaginal environment to pH < 4.5, and inhibit the growth of other viruses and bacteria [22]. Thus, *L. gasseri* is associated with women’s reproductive health [23]. Indeed, in our study, this species predominated in patients with ongoing pregnancy; however, it remains unclear how exactly the administration of a progestogen can modulate the growth of this bacterium.

*Bifidobacterium dentium*, according to our data, was statistically more represented in the group of patients with ongoing pregnancy who took dydrogesterone compared with the group where patients did not take it and the group of patients with miscarriage. This species has previously been found in the vaginal microbiota, but its role in women’s reproductive health remains unknown [24,25]. According to the research of Kato et al. (2022), *B. dentium* was significantly higher in the saliva of pregnant women than in non-pregnant women. In addition, the abundance of *B. dentium* in this study significantly correlated with the concentration of progesterone in saliva in all subjects [26]. In our study, we observed a similar effect in the cervical and vaginal environments, where *B. dentium* is abundant in the group of patients who are taking dydrogesterone, which is a synthetic progesterone drug. 

*Bifidobacterium longum* prevailed in patients with ongoing pregnancy who did not take dydrogesterone compared to the other two groups. However, we observed an inverse effect for *Bifidobacterium breve*, which dominated the vaginal samples of the miscarriage group compared to both ongoing pregnancy groups. *Bifidobacteria* are known to be generally considered beneficial members of the gut microbiota [27]. Although their role in the vaginal microbiota has not yet been elucidated, it can be hypothesized that lactic acid-producing *Bifidobacterium* may have a protective or health-promoting effect in the vagina similar to that of *Lactobacillus*. Early studies showed that *B. longum* produced the highest levels of lactic acid compared to other *Bifidobacterium* isolates. It is also known that most vaginal *Bifidobacterium* can tolerate high levels of lactic acid (100 mM) and low pH (4.5 or 3.9), characteristic of the vaginal secretion of healthy women [28]. Thus, studies show that *B. breve* and *B. longum* can be considered members of the bacterial community that contribute to reproductive health. The microbiota with *B. breve* and *B. longum* in patients in early pregnancy was associated with a reduced risk of premature spontaneous preterm birth [23,29]. However, we cannot confirm these findings for *B. breve* since, in our study, its abundance in the group of patients with miscarriage was statistically significantly higher than in the groups with ongoing pregnancy. Thus, our data highlight the lack of knowledge of *Bifidobacterium* species in the cervical and vaginal microbiomes. It also shows the need for further research to determine their impact on women’s reproductive health.

Statistically significant differences in the abundance of *Dialister invisus* were observed in samples collected from the cervical canal. The number of *Dialister invisus* was higher in the group of patients with ongoing pregnancy taking dydrogesterone compared to the group with miscarriages. There is little information in the literature about the role of this bacterium in female reproductive health. *Dialister invisus* is associated with early-term delivery in African American patients [30]. Also, one study showed that the abundance of *Dialister invisus* in the intestinal microbiota decreased in patients with bacterial vaginosis [31]. *Dialister invisus* was also significantly associated with the presence of HPV infection and the risk of high-grade squamous intraepithelial neoplasia and cervical cancer [32,33,34]. Thus, in our study, for the first time, an increase in the number of this species in a woman in early pregnancy taking dydrogesterone was shown. This finding highlights the need for further research into the effects of *Dialister invisus* on host health.

*Gardnerella vaginalis* is the dominant microorganism in the CST IVB microbiome type, according to the classification of France et al. (2020) [13]. We did not find significant differences in this type of microbiome between the three study groups. Meanwhile, the abundance of *G. vaginalis* shows significant differences between all three groups in both cervical and vaginal samples. The highest abundance of *G. vaginalis* was observed in the miscarriage group, followed by the ongoing pregnancy group without dydrogesterone usage, and the lowest abundance was observed in patients with dydrogesterone usage. In both cultural and non-cultural studies, *G. vaginalis* was associated with early and late miscarriages, as well as preterm birth [21,35,36]. It is known that the CST IVB microbiome type in the non-pregnant state is associated with a clinical picture of bacterial vaginosis, including higher pH, increased production of proinflammatory cytokines, such as interleukin IL-6 and IL-8, and human papillomavirus (HPV) progression [13,28,35,37]. Meanwhile, some studies showed that CST IV, including the dominance of *G. vaginalis*, is a common type of vaginal microbiome in African and Hispanic non-pregnant women [28], and these women also mostly have uncomplicated pregnancies [38,39]. There is some evidence that the abundance of *G. vaginalis* in the vaginal microbiome is higher in healthy women compared to patients with a history of miscarriage [40,41], but the limitations of both studies were a small sample size. Therefore, our findings confirm the important role of *G. vaginalis* in the pathogenesis of early miscarriages. Further studies are needed to solve some existing contraversions.

The highest number of *Mycoplasma girerdii* was found in the group of patients with miscarriage, compared with groups with ongoing pregnancy. It is known that other *Mycoplasma* species, in particular *Mycoplasma genitalium* and *Mycoplasma hominis*, can cause miscarriages and preterm births, irrespective of the presence of other sexually transmitted infections [42]. Despite that, there is still a discussion on the necessity to screen for and treat mycoplasma infections before and during pregnancy. *M. girerdii* is very commonly found in patients infected with *Trichomonas vaginalis* but occasionally occurs in the vaginal community of patients without *Trichomonas vaginalis* [43,44]. Early studies reported that *M. girerdii* was associated with an increased relative risk of bacterial vaginosis [44,45]. In addition, some studies have identified *M. girerdii* in several cohorts of the vaginal microbiome in preterm birth; however, given the low prevalence of the microorganism and the relatively small sample size in these studies, the association of the bacterium with preterm birth has yet to be adequately assessed [46,47,48]. In our study, there were no patients infected with *Trichomonas vaginalis*, which proves the possibility of the presence of *M. girerdii* in the microbiome of the female lower genital tract outside the context of trichomoniasis. In addition, our study shows for the first time the potential association of this bacterium with miscarriage, which indicates the need for further research on *M. girerdii* in the context of miscarriage. It can be suggested that screening for all *Mycoplasma* family species before pregnancy is necessary to avoid pregnancy complications. 

In the cervical microbiome, we observed a statistically significant difference in the abundance of *Prevotella copri*. This species predominated in the group of women with ongoing pregnancies who took dydrogesterone compared to the group where women did not take it. *Prevotella copri* is considered a normal member of the healthy vaginal microbiota. Some studies show a possible positive role for this bacterium, as it increased with probiotics and also predominated in women without bacterial vaginosis [31,49]. Our study also demonstrates the presence of this bacteria in the early stages of a healthy pregnancy, but it remains to be seen how progesterone supplementation may affect this bacterial species.

In our study, *Bacteroides plebeius* prevailed in the group of patients with miscarriage. This is the first study to show this association; previously, a high abundance of *Bacteroides plebeius* was associated with HPV infection [50].

## 4. Materials and Methods

### 4.1. Ethics and Patient Consent Guidelines

The study was conducted following the ethical standards of the Declaration of Helsinki by the World Medical Association (1964) and its subsequent amendments.

Written informed consent was obtained from all patients for the collection and usage of their biomaterial, as well as data of the anamnesis and clinical examination, including the publication of the data obtained after the anonymization. The study was approved by the Ethical Committee of Voronezh State University (protocol No. 42-05 of 27 December 2021).

### 4.2. Study Design

In a prospective longitudinal cohort study, pregnant women (*n* = 51) were recruited at the beginning of their antenatal care (8–11 weeks of gestation) at Yakovlevo Central District Hospital (Belgorod region, Russia) from January 2022 to January 2023. Patients were divided into three groups: women with ongoing pregnancy without the usage of dydrogesterone (group I, *n* = 23), patients with ongoing pregnancy with the usage of dydrogesterone 10 mg twice a day per os for at least three weeks (group II, *n* = 17), and women with miscarriages (group III, *n* = 11). Only patients with an ongoing pregnancy, delivered at term (37 to 42 weeks of gestation), were included in the first and second groups. The indications for the dydrogesterone administration were a history of recurrent pregnancy loss, threatened miscarriage during the current pregnancy, and a history of infertility due to luteal insufficiency. The diagnosis of miscarriage in the third group was made based on clinical examination and ultrasound examination and then confirmed by histopathological examination. See Appendix A for complete patient information. 

Exclusion criteria for all groups included vaginal bleeding, the usage of any antibiotics, probiotics, prebiotics, or synbiotics during the previous month, vaginal intercourse during the last 3 days, the presence of severe non-obstetrical conditions, including primary and secondary immune deficiency, uterine malformations, fetal chromosomal abnormalities, refusal to sign the informed consent form, or refusal to be followed up. Basic clinical characteristics are shown in Table 1.

To study the microbiome of the female genital tract, the material was collected using a cytobrush (cervical canal) and a vaginal swab (vagina). The biomaterial was collected in two Eppendorf tubes (the 1st from the cervical canal by a cytobrush; the 2nd from the vagina by a vaginal swab) with RNAlater™ stabilization solution (Thermo Fisher Scientific, Madison, WI, USA), delivered to the laboratory, and stored at +4 °C for 24 h according to the instructions. After 24 h, the biomaterial was stored at a temperature of −80 °C until the start of the investigation.

### 4.3. Microbial DNA Extraction and Next-Generation Sequencing

We used the ZymoBIOMICS DNA Miniprep Kit for total DNA isolation. The DNA extraction procedure was performed from cervical and vaginal swabs according to the manufacturer’s protocol (Zymo Research, Los Angeles, CA, USA). During the DNA isolation step, an extra sample containing Milli-Q laboratory water and a clipped, sterile cytobrush was added. Further, this control sample participated in sample preparation in the same way as the testing samples. This step is necessary so that, at the stage of bioinformatics data analysis, we can improve the accuracy of our microbiome data by removing contaminants present in the laboratory. Quantitative control of isolated DNA was performed using a Qubit 2.0 benchtop fluorometer with a double-stranded DNA analysis kit (Thermo Fisher Scientific, Madison, WI, USA). The concentration of all studied samples was acceptable for further preparation of libraries for sequencing.

For our study, we chose the strategy of targeted sequencing of the V3 hypervariable region of the *16S rRNA* gene. For targeted amplification of this region, we used the 337F and 518R primer sets in combination with the 5X ScreenMix-HS Master Mix Kit (Evrogen, Moscow, Russia) for amplification on a Bio-Rad CFX96 Touch™ instrument (Bio-Rad, Hercules, CA, USA). The amplification protocol included the main stages: total denaturation at 94 °C for 4 min, followed by 25 cycles consisting of denaturation at 94 °C for 30 s, primer annealing at 53 °C for 30 s, and elongation at 72 °C for 30 s. The final elongation lasted 5 min at a temperature of 72 °C.

To prepare sequencing libraries for the Ion Torrent PGM platform, *16S rRNA* gene amplification products were purified from reaction components using AMPureXP magnetic particles (BeckmanCoulter, Brea, CA, USA). Next, a commercial NEBNext Fast DNA kit (New England Biolabs, Ipswich, MA, USA) was used to process the ends of the amplicons and ligate barcodes with adapters according to the manufacturer’s protocol. The libraries were barcoded with a NEXTflex Kit 1-64 (PerkinElmer, Inc., Waltham, MA, USA). The barcoded libraries were again purified with AMPureXP magnetic particles, adding particles at a ratio of 1.1 by sample volume (Beckman Coulter, Brea, CA, USA). The last step in library preparation was concentration measurement using the KAPA Library Quantification Kit Ion Torrent™ Platforms (F. Hoffmann-La Roche AG, Basel, Switzerland).

The Ion PGM Hi-Q View OT2 kit was used to enrich the emulsion polymerase chain reaction (PCR) libraries; the Ion PGM Hi-Q View kit and the Ion 318™ Chip v2 BC chip (ThermoFisher Scientific, Madison, WI, USA) were used to run the instrument and the sequencing reaction.

### 4.4. Bioinformatics and Statistical Analysis

Sequencing data were preprocessed based on the Ion Torrent Server (ThermoFisher Scientific, Madison, WI, USA). Also, BAM files were generated for each sample, which, for further work, we converted to the FastQ format using the built-in FileExporter plugin. We performed further data analysis in the R Studio environment (V SEARCH v software 2.8.2; RStudio Inc., Boston, MA, USA). All low-quality reads for which the maximum predicted error threshold criterion was less than 1.0 were removed from further analysis (DADA2 package). Then, all reads were trimmed to universal length and demultiplexed. During dereplication, we combined all identical reads into unique sequences, and based on the UNOISE2 algorithm, we identified operational taxonomic units (OTUs).

Taxonomic assignment was carried out using the SILVA database (https://www.arb-silva.de/, accessed 15 July 2023). The relative threshold for peak similarity with variant amplicon sequences was set at 97%.

We used the Shannon index to evaluate the alpha diversity of the studied samples. The statistical significance of the observed differences was assessed using the Wilcoxon rank sum test adjusted by the false discovery rate (FDR) method. Fisher’s exact test estimated differences between CSTs in groups. The intergroup similarity of the studied microbiomes was assessed using nonmetric multivariate scaling (NMDS) using the Bray–Curtis dissimilarity according to the ANOSIM test. Differential abundance analysis was performed in GraphPad Prism 9 software (GraphPad, San Diego, CA, USA) using the Multiple Wilcoxon test.

All raw sequences are available from the BioProject repository (Bio Project: PRJNA886610).

## 5. Conclusions

In the study, we estimated the microbiome composition of patients with early miscarriages, ongoing pregnancy without medical support, and ongoing pregnancy with dydrogesterone support, which can be considered a “near-missed pregnancy”. The data we obtained demonstrate that *L. iners* is a predominant microorganism in eastern European women’s cervical and vaginal microbiomes during the first trimester of ongoing pregnancies with or without dydrogesterone usage and miscarriages. We did not find any significant differences in CSTs between the study groups. Both cervical and vaginal microbiomes of patients with miscarriage were the richest and most diverse compared to those of patients with ongoing pregnancy with and without progesterone usage. 

In our research, the cervical microbiome was taken by a cytobrush; therefore, it contained intracellular bacteria. We found a significant decrease in the abundance of *Bifidobacterium dentium* and *Dialister invisus* in patients with miscarriages compared to patients with ongoing pregnancy using dydrogesterone, an increase in the abundance of *Gardnerella vaginalis* and *M. girerdii*, and a decrease in the abundance of *Bifidobacterium longum* compared to all cases of ongoing pregnancy. The abundance of *L. iners* was significantly higher in women with ongoing pregnancy without dydrogesterone usage compared to the other two groups. 

In the vaginal microbiome, patients with miscarriages had a significantly higher abundance of *Gardnerella vaginalis*, *M. girerdii*, and *Bifidobacterium breve* and a lower abundance of *Bifidobacterium longum* compared to all cases of ongoing pregnancy, as well as a higher abundance of *Bacteroides plebeius* compared to patients with ongoing pregnancy without dydrogesterone treatment. The highest abundance of *L. iners* was observed in patients with ongoing pregnancy without progesterone usage, followed by ongoing pregnancy with progesterone usage and miscarriages. The abundance of *Bifidobacterium dentium* and *Lactobacterium gasseri* was significantly higher in women with ongoing pregnancy with dydrogesterone treatment compared to the other two groups.

The data obtained showed the important role of *Gardnerella vaginalis* and *M. girerdii* in the pathogenesis of early miscarriages and the necessity of screening for these infections during preconception care. *L. iners* and *Bifidobacterium longum* have a protective effect during early pregnancy. We found significant differences between the cervical and vaginal microbiomes depending on dydrogesterone intake. Further investigations are needed to define the causes of these differences. 

The study contributes to understanding the cervical and vaginal microbiome composition of the eastern European population in early pregnancy depending on progesterone treatment, as well as the role of the genital tract microbiome in the pathogenesis of early miscarriages.

## Figures and Tables

**Figure 1 ijms-24-13836-f001:**
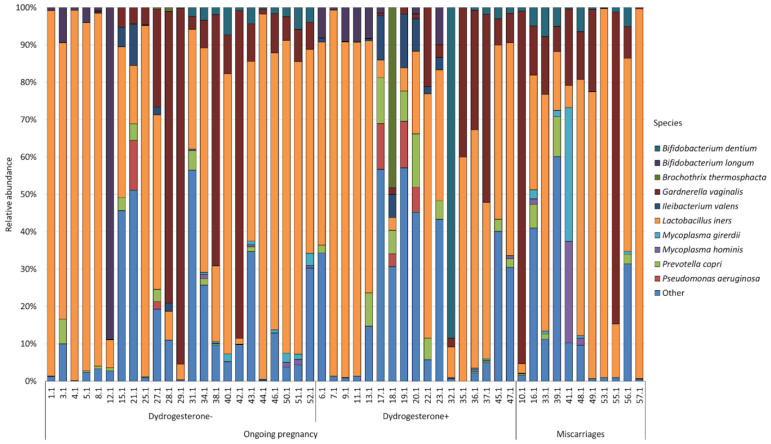
Main composition of the cervical microbiota in the group of patients with ongoing pregnancy not taking dydrogesterone and taking dydrogesterone and patients with miscarriage.

**Figure 2 ijms-24-13836-f002:**
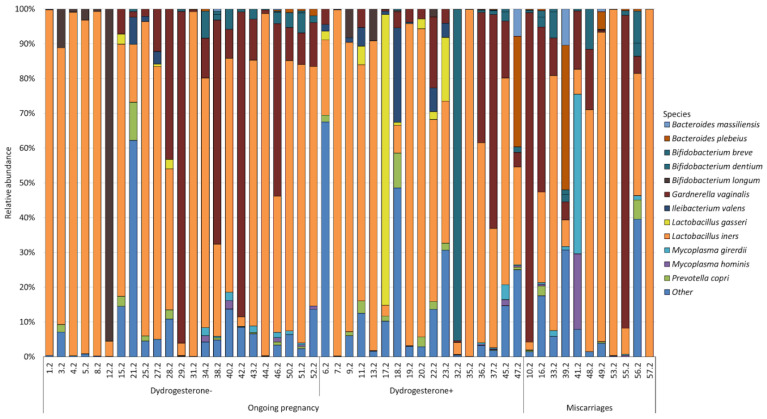
Main composition of the vaginal microbiota in the group of patients with ongoing pregnancy not taking dydrogesterone and taking dydrogesterone and patients with miscarriage.

**Figure 3 ijms-24-13836-f003:**
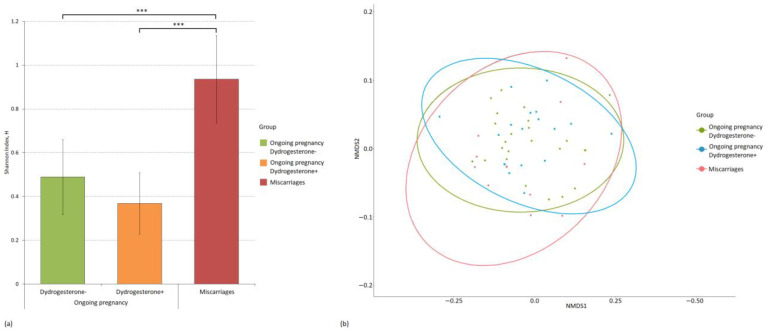
Evaluation of intragroup and intergroup indicators of the diversity of the microbiome of the cervical canal of patients divided into study groups. (**a**) Alpha diversity of the microbiome of study groups using the Shannon index and *p-values* (*** *p-value* ≤ 0.001). (**b**) Principal coordinate analysis (PCoA) plot of beta diversity based on the Bray–Curtis dissimilarity index between microbial community samples in study groups.

**Figure 4 ijms-24-13836-f004:**
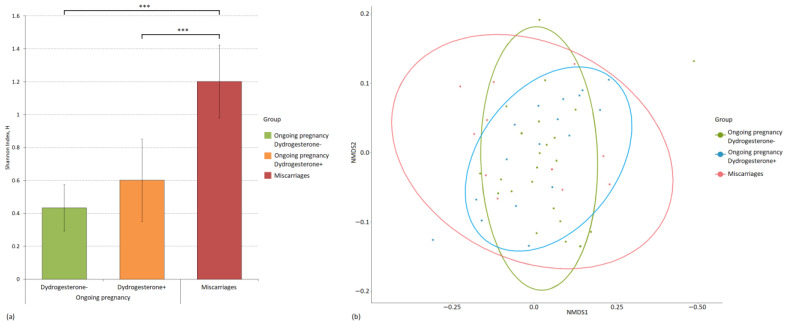
Evaluation of intragroup and intergroup indicators of the diversity of the microbiome of the vagina of patients, divided into study groups. (**a**) Alpha diversity of the microbiome of study groups using the Shannon index and *p-values* (*** *p-value* ≤ 0.001). (**b**) Principal coordinate analysis (PCoA) plot of beta diversity based on the Bray–Curtis dissimilarity index between microbial community samples in study groups.

**Figure 5 ijms-24-13836-f005:**
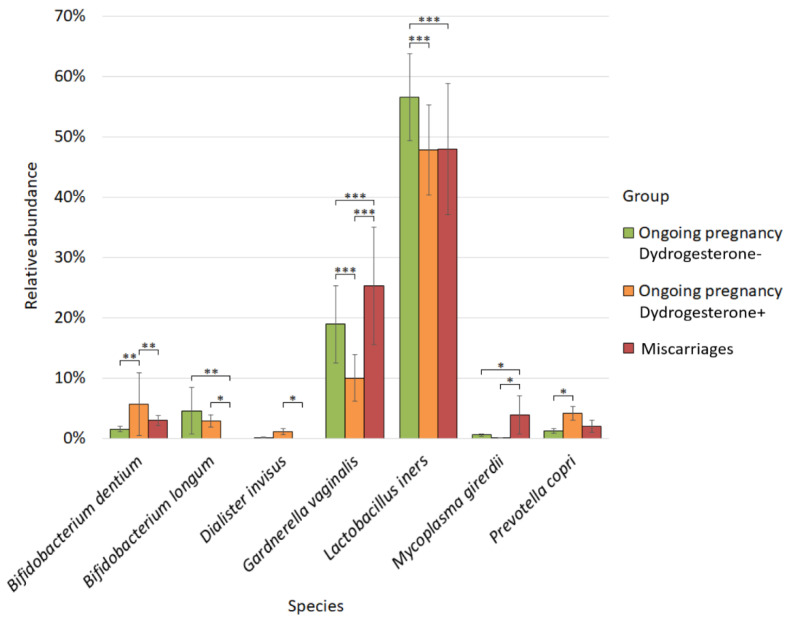
Differences in the cervix microbiome composition between studied groups. * *p-value* ≤ 0.05, ** *p-value* ≤ 0.01, *** *p-value* ≤ 0.001.

**Figure 6 ijms-24-13836-f006:**
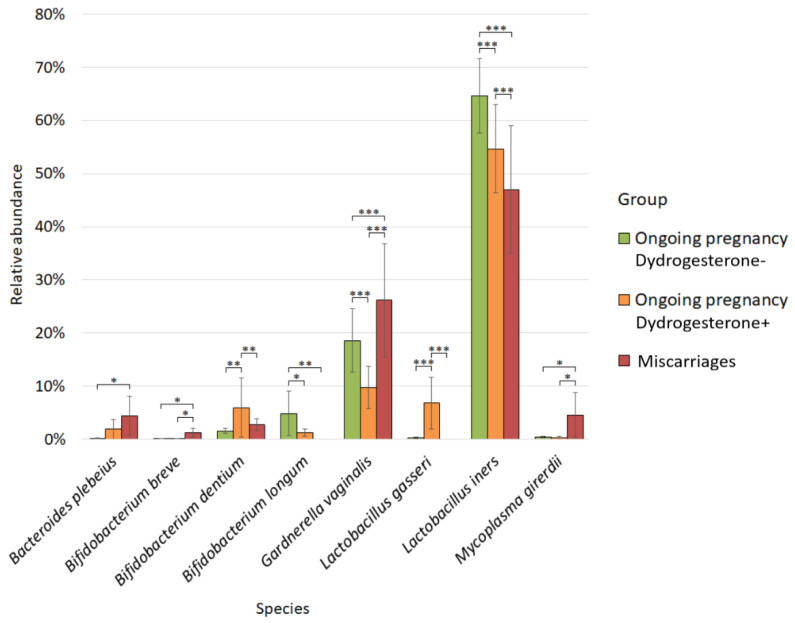
Differences in the vaginal microbiome composition between studied groups. * *p-value* ≤ 0.05, ** *p-value* ≤ 0.01, *** *p-value* ≤ 0.001.

**Table 1 ijms-24-13836-t001:** Clinical characteristics of patients enrolled in the study (*n* = 51) (M ± SEM).

Parameter	Group I (Ongoing Pregnancy without Dydrogesterone, *n* = 23)	Group II (Ongoing Pregnancy with Dydrogesterone, *n* = 17)	Group III (Miscarriages, *n* = 11)
Age (years)	28.87 ± 1.34	31.88 ± 1.18	28.27 ± 1.83
Gravidity	1.44 ± 0.35	1.88 ± 0.31	2.09 ± 0.62
Parity	0.96 ± 0.24	0.88 ± 0.19	1.00 ± 0.33
Number of miscarriages in anamnesis	0.13 ± 0.07	0.59 ± 0.17 **	0.73 ± 0.31 *
Number of artificial abortions in anamnesis	0.35 ± 0.14	0.35 ± 0.15	0.36 ± 0.20
Weight	66.03 ± 3.13	64.98 ± 2.66	61.73 ± 3.10
Hight	165.4 ± 1.49	165.4 ± 1.53	163.3 ± 0.82
Body mass index (BMI)	24.03 ± 1.01	23.73 ± 0.75	23.14 ± 1.11

* *p*_I–III_ < 0.05, ** *p*_I–II_ < 0.01.

## Data Availability

The raw sequencing data are available in the NCBI BioProject database (BioProjectID: PRJNA886610).

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
