# Peer review of "Cervical and Vaginal Microbiomes in Early Miscarriages and Ongoing Pregnancy with and without Dydrogesterone Usage"

_ijms, 2023, doi:10.3390/ijms241813836_

Round 1

Reviewer 1 Report

Point 1:- graphical abstract is required 

Point 2:- In each groups the number of patients are different , do that affect the results 

Author Response

Dear reviewer!

We you would like to thank you for the review.

Here are our notes concerning review points:  

Reviewer point #1.1: graphical abstract is required

Author response #1.1: Graphical abstract is added (please, see in attached file).

Reviewer point #1.2: In each groups the number of patients are different, do that affect the results

Author response #1.2: As we had non-Gaussian (no-normal distribution) in groups, we used Mann-Whitney U-test for comparison of groups, because groups are not pair-wise. Mann-Whitey U-test can be used for unequal sample-size groups

“Wilcoxon (1945)  first proposed the test for equal sample sizes, and then Mann & Whitney (1947)  extended the test to cover different sample sizes”. https://influentialpoints.com/Training/Wilcoxon-Mann-Whitney_U_test_use_and_misuse.htm#:~:text=Wilcoxon%20(1945)%20first%20proposed%20the,to%20cover%20different%20sample%20sizes

Reviewer 2 Report

This work estimated the microbiome composition of patients with early miscarriages, ongoing pregnancy without medical support, and ongoing pregnancy with dydrogesterone support. The data demonstrated that both cervical and vaginal microbiomes of patients with miscarriage were the richest and most diverse compared to patients with ongoing pregnancy with or without progesterone usage. This is a very meaningful work. However, the following issues should be addressed before the paper is considered suitable for publication in International Journal of Molecular Sciences.

1.      There are numerous formatting errors in the references, please correct them promptly. For example, ref. 16 and ref. 46.

2.      Many of the words in the text are abbreviated, but not fully written. Please add full names for them, such as NGS, CST, and FDR.

3.      Please indicate the error range (p-value) in the diagram. For example, figures 5 and 6.

4.      The region mentioned in this paper has a certain influence on the distribution of microbiome in the vagina of pregnant women, but only 51 pregnant women in Russia were studied in this paper. Whether it is representative or not, please supplement the relevant explanation.

5.      Introduction, the author mentions “Differences in microbiome composition persist during normal early pregnancy”. In order to support this statement, the following recently published important related papers should be cited: Exploration 2021, 1, 21; Chem. Soc. Rev. 2021, 50, 2839; VIEW. 2023;20220064; Adv. Mater. 2023, DOI: 10.1002/adma.202304249.

Extensive editing of English language required

Author Response

Dear reviewer!

We you would like to thank you for the review.

Here are our notes concerning review points:  

Reviewer point #2.1: There are numerous formatting errors in the references, please correct them promptly. For example, ref. 16 and ref. 46.

Author response #2.1: Thank you for your comment, we understood what do you mean and double-checked all our references. When compared with the source, indeed, the names of the pages in some cases do not look like 1-15, but as in the case of link 46: 356.e1-356.e18 it is correct. (https://www.ncbi.nlm.nih.gov/pmc/articles/PMC5581228/#:~:text=Within%20subject%20comparisons%20across%20pregnancy,were%20associated%20with%20preterm%20birth)

Reviewer point #2.2: Many of the words in the text are abbreviated, but not fully written. Please add full names for them, such as NGS, CST, and FDR.

Author response #2.2: Thank you for your comment, we have added all full terms to the text.

Reviewer point #2.3: Please indicate the error range (p-value) in the diagram. For example, figures 5 and 6.

Author response #2.3: Thank you for noticing the lack of p-value in the corresponding figures. We have added p-value information to the description of the figures.

Reviewer point #2.4: The region mentioned in this paper has a certain influence on the distribution of microbiome in the vagina of pregnant women, but only 51 pregnant women in Russia were studied in this paper. Whether it is representative or not, please supplement the relevant explanation.

Author response #2.4: Our research was supported by the Russian Science Foundation as a pilot study, as cervical and vaginal microbiome in early pregnancy in Russia have not been studied yet in ongoing pregnancy and miscarriages both. In the materials and methods section, we indicate that our sample consists of women of Russian nationality living in the Belgorod region, which belongs to the central part of Russia. It is planned to extend this research in larger cohorts next year. Similar research were made with similar or less amounts of samples, please, see below:

  1. Huang, Y.; Li, D.; Cai, W.; Zhu, H.; Shane, M.I.; Liao, C.; Pan, S. Distribution of Vaginal and Gut Microbiome in Advanced Maternal Age. Front. Cell. Infect. Microbiol. 2022, 12, 819802.
  2. Zheng, N.; Guo, R.; Yao, Y.; Jin, M.; Cheng, Y.; Ling, Z. Lactobacillus Iners Is Associated with Vaginal Dysbiosis in Healthy Pregnant Women: A Preliminary Study. Biomed Res Int 2019, 2019, 6079734, doi:10.1155/2019/6079734.
  3. Huang, Y.-E.; Wang, Y.; He, Y.; Ji, Y.; Wang, L.-P.; Sheng, H.-F.; Zhang, M.; Huang, Q.-T.; Zhang, D.-J.; Wu, J.-J.; et al. Homogeneity of the Vaginal Microbiome at the Cervix, Posterior Fornix, and Vaginal Canal in Pregnant Chinese Women. Microb Ecol 2015, 69, 407–414, doi:10.1007/s00248-014-0487-1.

Reviewer point #2.5: Introduction, the author mentions “Differences in microbiome composition persist during normal early pregnancy”. In order to support this statement, the following recently published important related papers should be cited: Exploration 2021, 1, 21; Chem. Soc. Rev. 2021, 50, 2839; VIEW. 2023;20220064; Adv. Mater. 2023, DOI: 10.1002/adma.202304249.

Author response #2.5: We would like to thank the reviewer for advising such interesting article. Meanwhile, this article provides information on nanoparticles use in cancer, which has no relation to early pregnancy and genital tract microbiome. Therefore, we cannot cite this article in our manuscript.

Authors response on English language editing: One of co-authors, Prof. Alexander Maslov, who has been working  at the university in United States for 22 years, revised manuscript again, some changes has been done by him, mostly in the abstract of the article. Please, see these changes in the manuscript.

Reviewer 3 Report

No comments, well-written manus

Author Response

Dear reviewer! 

We would like to thank you for the review.